# Synergetic Improvement of Interfacial Performance and Impact Resistance of Carbon Fiber-Reinforced Epoxy Composite via Continuous Electrochemical Oxidation

**DOI:** 10.3390/polym17081007

**Published:** 2025-04-08

**Authors:** Ziqi Duan, Weidong Li, Hansong Liu, Pengfei Shen, Huanzhi Yang, Xiangyu Zhong, Jianwen Bao

**Affiliations:** National Key Laboratory of Advanced Composites, AVIC Composite Technology Center, AVIC Composite Corporation Ltd., Beijing 101300, China; dzq970920@163.com (Z.D.); lhsarticle@163.com (H.L.); 19821277882@163.com (P.S.); huanzhiyangavic@163.com (H.Y.); zxyarticle@163.com (X.Z.)

**Keywords:** carbon fiber, continuous electrochemical oxidation, interfacial performance, low-speed impact

## Abstract

This study elucidated the synergistic improvement of the interfacial performance and impact resistance of carbon fiber-reinforced epoxy (CF/EP) composites by employing continuous electrochemical oxidation. CFs were electrochemically treated at different current densities, and the surface physicochemical properties, including the surface morphologies, chemical compositions, and wettabilities, were analyzed. After electrochemical oxidation, the IFSS of the CF/EP increased by 48.8%, significantly improving the impact resistance of the composites. The experimental results showed that, when the current density reached 0.15 mA/cm^2^, the damage area after impact reduced by 61%. Concurrently, fiber fracture and resin failure became the primary energy-dissipation modes, maximizing the fiber reinforcement effect and enhancing the impact resistance. However, fiber fracture deteriorated the static mechanical properties of the composites. Subsequently, at a 0.10 mA/cm^2^ current density, the CF/EP composites exhibited an increased compressive strength after an impact of 331 MPa.

## 1. Introduction

Carbon fibers (CFs) exhibit several desirable properties, such as a high specific modulus, specific strength, heat resistance, and corrosion resistance. As a result, they are key strategic materials in the aerospace sector and form the primary reinforcement materials for advanced composites [1,2,3]. The compressive strength after impact (CAI) of third-generation high-toughness epoxy (EP) composites reinforced by high-strength intermediate-modulus carbon fibers (IMCFs)—namely, M91/IM7, M21E/IMA, and 3900–2/T800H—reaches 315 MPa. Therefore, they have been widely used in the primary load-bearing structures of aircraft [4,5,6]. The utilization of advanced composites has become a notable indicator of the technological advancement in current aircraft [7,8]. The resistance to low-speed impact remains a performance bottleneck, limiting the application of composites in the aerospace sector [8].

The strengths and moduli of IMCFs are higher than those of high-strength CFs. However, the anisotropy of their composites is more prominent, resulting in a greater susceptibility to delamination under impact loads. Therefore, improving the impact resistance of IMCF-reinforced composites is a key factor in determining whether IMCF can significantly reduce the weight to a greater extent [9,10]. The impact resistance of a composite is generally considered to be related to its matrix toughness, interfacial properties, and interlayer toughness [11]. The interfacial bonding between CF and resin plays an important role in the anti-delamination performance of composites. In the case of IMCFs, owing to the higher temperature of carbonization treatment, the CF surface exhibits stronger chemical inertness. This results in poor interfacial bonding between IMCF and resin, which limits the improvement of the impact resistance [8]. Various surface treatment techniques to enhance interfacial bonding and improve the performance of composites have been widely studied, for example, electrochemical oxidation [12,13,14,15,16,17,18,19,20,21], γ-ray irradiation [22], chemical grafting [23,24], vapor deposition treatment [25], silane coupling agent coating [5,26], plasma treatment [27,28], and the application of nanoparticles [29,30]. Electrochemical oxidation is particularly preferred in the industrial production of CFs [31]. The electrochemical oxidation method exhibits many advantages, including a mild reaction, short processing time, uniform oxidation, and the precise control of the oxidation degree, which can improve the surface wettability and reactivity of CFs. This, in turn, significantly enhances the CF/EP interfacial properties, as well as the mechanical properties. In addition, electrochemical oxidation can be integrated with continuous production lines to achieve the large-scale online surface modification of CFs.

As regards electrochemical oxidation, numerous prior studies have examined high-strength high-modulus CFs [12,13,16,17,18,19,20,21], focusing on electrolyte types and concentrations, electrochemical treatment parameters (e.g., temperature, treatment time, and current density), and mechanisms. Yumitori et al. [14,15] investigated the effects of several electrolytes on oxygen-containing functional groups on the CF surface and the interfacial strength of composites. Cao et al. [17] compared the effects of two types of ammonium phosphate salts as electrolytes on the degree of surface etching and interlayer shear strength (ILSS) of composites. Liu et al. [13] and Zhang et al. [32] studied the microstructure of CFs by using complex electrolytes and assessed the ILSSs of the composites. Ji et al. [33] used NH_4_HCO_3_ to electrochemically oxidize the surface of T300-grade CF and investigated the optimal process parameters. The interlayer shear strength of CF-reinforced thermoset resin reached 90 MPa at an electrolyte mass fraction of 9.5%, a current density of 1.2 mA/cm^2^, an oxidation time of 90 s, and a temperature of 34 °C. Qiao [34] treated M55J with NH_4_HSO_4_ electrolyte by employing a 6.5 wt% electrolyte concentration and 1.0 mA/cm^2^ current density. The ILSS of the M55J/EP composite reached 66 MPa. Mittal et al. [35] used 5 wt% KMnO_4_, concentrated HNO_3_, and [(NH_4_)_2_HPO_4_] as electrolytes to electrochemically treat CFs and determined that the optimal treatment times for the oxidation treatment were 10 min, 4 h, and 1 h, respectively. Sun et al. [36] used NH_4_HCO_3_ to oxidize 6K T700-grade CF and determined the following optimal process conditions: an electrolyte concentration of 5 wt%, a treatment time of 80 s, a current intensity of 40 mA/cm^2^, and a temperature of 55 °C.

The electrochemical oxidation mechanisms of CF surfaces have been widely studied, and various electrochemical oxidation models have been proposed. Fukunaga et al. [16] reported that the electrochemical modification of CF preferentially involves oxidation reactions at cracks or depressions, resulting in the formation of a prismatic surface at the grain boundary edges. Liu et al. [37] explored the electrochemical surface modification mechanism of CF and reported that the CF underwent weak layer generation, morphology evolution, and morphology alteration stages during electrochemical surface modification. Bismarck et al. [38] studied the cyclic voltammetry curves of high-modulus CFs after electrochemical oxidation in a KClO_4_/KOH complex electrolyte and proposed an electrochemical modification mechanism in alkaline electrolytes. Guo et al. [39] examined the mechanism underlying CF surface modification by electrochemical oxidation and proposed a “physicochemical dual-effect” mechanism model to improve the physical and chemical states of the CF surface.

The electrochemical oxidation method exhibits many advantages and has broad prospects for the industrial production of CF. However, the current research remains in the laboratory stage, disconnected from the application of CF-reinforced composites. Meanwhile, existing studies have focused more on the effects of electrochemical oxidation on surface of CFs and the microinterface between CFs and resins, neglecting the key performance in the application of composites—impact damage resistance. Consequently, the research findings have been inadequate in meeting the requirements for the practical development of the composites.

With a focus on industrial production, this study used a continuous electrochemical oxidation method to treat IMCFs in a large-scale production line. Abundant physical, chemical, and mechanical characterization techniques at both the micro- and macroscales were used to examine the effects of the current density on the physiochemical properties of IMCFs, the CF/EP interfacial bonding strength, and, especially, on the damage resistance and compression properties after the impact of IMCF-reinforced composites qualitatively and quantitatively to provide a basis for the application of IMCF-reinforced composites.

## 2. Materials and Methods

### 2.1. Materials

This study used T800H-grade CF (Weihai Tuozhan Fiber Co., Ltd., Weihai, China). Table 1 presents the basic properties of the CF tested according to ASTM D4018 [40] or GB/T 3362-2017 [41]. Before the microscale characterization, CF desizing was performed using the Soxhlet method. The EP resin was obtained from AVIC Composite Corporation Ltd. (Beijing, China).

### 2.2. CF Preparation Using Different Electrochemical Treatment Parameters

The CF production process entails preoxidation, carbonization, surface treatment (electrochemical treatment and sizing), and wire winding. In the electrochemical treatment process (Figure 1), the electrolysis temperature and time were 30 °C and 70 s, respectively, with NH_4_HCO_3_ solution (10 wt%) used as the electrolyte. The current densities considered were 0, 0.05, 0.10, 0.15, and 0.20 mA/cm^2^, and the obtained CFs were designated as CF-0, CF-1, CF-2, CF-3, and CF-4, respectively. The information of the CF samples are shown in Table 2.

### 2.3. Preparation of CF/EP Laminates

The CF/EP prepreg was prepared with a fiber area density of (145 ± 3) g/m^2^ and a resin content of 35% via the two-step hot-melt method. The prepreg was laid in the sequence [45/0/−45/90]_4S_ according to ASTM D7136 (America Standard) [42] and GB/T 21239-2022 (China National Standard) [43], and the CF/EP laminates were manufactured using the autoclave molding method, conforming to the curing process depicted in Figure 2. The cured laminates were cut into impact samples (150 mm × 100 mm × 4.48 mm). The CF laminates had a CF volume fraction of 57%.

#### 2.3.1. Tests for Surface Physicochemical Properties

The morphologies of the CF surfaces subjected to different electrochemical treatments were examined using scanning electron microscopy (SEM, JEOL JSM-6010, Tokyo, Japan) and atomic force microscopy (AFM, Veeco DI Innova, New York, NY, USA). The roughnesses of the CF surfaces were ascertained using NanoScope Analysis 3.0. To characterize the chemical properties of the CF surfaces, while the elemental components and active functional groups were investigated using an X-ray photoelectron spectroscopy analyzer (XPS, Thermo Fisher EscaLab 220i–XL, Thermo Fisher, Boston, MA, USA). The contact angles between the CFs and test liquids, including deionized water and ethylene glycol, were measured using a dynamic contact angle test meter (DCA, Biolin Scientific Sigma 702, Gothenburg, Sweden), and the surface energy of the CFs was calculated using the Owens, Wendt, Rabel, and Kaelble (OWRK) method. For all of the experiments, three monofilaments from different CF samples were tested.

#### 2.3.2. Monofilament Tensile Tests

Monofilament tensile testing was performed to investigate the influence of electrochemical treatment on the mechanical properties of the CFs. To this end, a mechanical testing machine (Instron 3342, Instron, Boston, MA, USA) was employed. According to ISO 11566-1996 (International Standard) [44] or GB/T 31290 (China National Standard) [45], both sides of a single CF filament were attached onto a backing paper. The distance between two bonding points was 25 mm. Before commencing the tests, the backing paper was snipped. Each CF sample was tested at least 30 times at a loading speed of 2 mm/min. Finally, a bivariate Weibull distribution was used to calculate the tensile strength of each filament.

#### 2.3.3. Interfacial Shear Strength Tests

The influence of the electrochemical treatment on the CF/EP interfacial properties was examined via the interfacial shear strength (IFSS) test using a self-made microsphere debonding device. The EP resin was dripped onto CF monofilaments and cured at 180 °C for 3 h, which is the curing cycle of the EP in this study. This yielded resin microspheres, which were subsequently fixed between a pair of blades before being tested. Then, the CF monofilament was pulled at 0.05 mm/min to peel the microsphere off the CF surface. During testing, the maximum force (F_max_), CF diameter (d), and length of the CF coated by resin microspheres (l) were recorded, and the IFSS was calculated as follows [46]:(1)$$IFSS=\frac{F_{max}}{\pi dl}.$$

Each type of CF was tested at least 10 times.

#### 2.3.4. Low-Speed Impact and Compression Tests

A low-speed impact test for the CF/EP composite was performed using a drop hammer impact testing machine (Instron9250HV, Instron, Boston, MA, USA), in accordance with ASTM D7136 [42], with an impact energy of 6.67 J/mm. The diameter and total mass of the hammer were 16 mm and 5.5 kg, respectively. At least five sets of samples were tested for each CF/EP group. The ultrasonic C-scan method was used for nondestructive testing to determine the damaged area generated within the samples after impact. The samples were cut at the impact location, and the cut surfaces were polished using sandpaper and diamond polishing powder. An optical microscope was then used to directly observe the cut fractures. After the impact and ultrasonic C-scan, the samples were subjected to compression tests by using a universal mechanical testing machine (Instron5982, Instron, Boston, MA, USA), in accordance with ASTM D7137 (America Standard) [47] or GB/T 21239-2022 [43].

## 3. Results and Discussion

### 3.1. Surface Morphology

To characterize the effect of current density on the CF surface morphology, SEM and AFM were performed. The results are depicted in Figure 3 and Figure 4, respectively. Figure 3 shows grooves of varying depths on the CF surfaces, parallel to the axis direction. These are attributable to the wet-spinning method. As the current density increased, the grooves on CF surfaces did not widen or deepen. However, electrochemical etching occurred. The surface roughness at various current densities did not show significant differences (Figure 4 and Table 3), and this observation is consistent with the SEM results. The groove structure enhanced the anchoring effect between the CF and matrix resin, thereby bringing benefit to the interfacial properties of the composites. With an increasing current density, charges accumulated at the grooves, forming charge concentration points, which intensified the electrochemical reactions and resulted in the oxidation etching phenomenon [37]. Therefore, oxidation and electrochemical etching are two reaction mechanisms in the electrochemical anodizing process. The surface inertness of the CF increased owing to the etching, thus lowering the interfacial strength of the composites.

### 3.2. Chemical Characteristics

The correlation between the current density and elemental composition of the CF surfaces was examined via XPS. The relative contents of C, N, and O on the CF surfaces are listed in Table 4. Figure 5a shows the wide-scan XPS spectra of the CF surfaces. With an increasing current density, the intensity of O1s on the CF surface increased, whereas the intensity of C1s slightly decreased. As current density increased, the number of active oxygen atoms generated by the electrolytic reaction increased. The contact between active oxygen atoms and active carbon atoms on the CF surfaces increased. This led to the production of more oxygen-containing functional groups on the CF surfaces, leading to a significant increase in the ratio of O atoms to C atoms from 0.084 to 0.173 [31]. Simultaneously, ammonia molecules from NH_4_HCO_3_ continuously reacted on the CF surfaces, which generated amino and imino groups, resulting in more nitrogen-containing functional groups and an increased N1s peak intensity [34].

The C1s peaks of the XPS spectra at different current densities were fitted and analyzed. The results are presented in Table 5. As the current density increased, the oxygen-containing functional group (–OH and –C=O) content on the CF surface increased to varying degrees. When the current density was lower than 0.15 mA/cm^2^, the –OH content remarkably increased from 15.37% to 24.14%, and the –C=O content slightly increased from 5.25% to 6.48%. However, as the current density exceeded 0.15 mA/cm^2^, the –OH content slightly decreased to 21.84%. On the contrary, the –C=O content continued to exhibit an increasing trend. As the intensity of the electrolytic treatment increased, five types of reactions occurred on the CF surfaces [37]. The first was the oxidation of carbon atoms on the CF surface, which led to an increase in the –OH content. Second, the existing and newly generated –OH groups were further oxidized to form –C=O. In the third type, –C=O generated CO_2_ under severe oxidation conditions, resulting in the removal of oxygen, which is called electrochemical etching. The fourth type was the reaction between ammonia molecules and active carbon atoms on the CF surfaces, leading to the generation of amino and imino groups. The last type was the formation of –C=O and N_2_ from the pyridine structure on the CF surfaces under severe electrolytic treatment. The first reaction was prominent when the current density was below 0.15 mA/cm^2^. As the current density increased, the second, third, and fifth reactions became increasingly prominent. The combined effect of these reactions caused a slight decrease in the –OH content and a significant increase in –C=O.

In summary, when the current density was increased to 0.15 mA/cm^2^, the nitrogen and oxygen contents reached their maximum values, increasing 2.11 and 0.88 times, respectively. Particularly, the carbon content reached the minimum, which was lower than that of CF-0 by 8.65%. The change in the oxygen content was determined by the synergistic effect of the aforementioned first, second, third, and fifth reactions. Conversely, the nitrogen content varied because of the synergistic effect of the fourth and fifth reactions. The –OH and –C=O on the CF surface aided the enhancement of the chemical bonding between the CF and resin, thereby benefitting the interfacial strength of the composites. Nitrogen-containing functional groups, such as amino and imine groups, can also undergo chemical reactions with the matrix resin to form chemical bonds, thus improving the interfacial properties.

### 3.3. Wettability of CF

The wetting process between the CF surface and matrix resin plays a crucial role in interfacial adhesion within composites. To quantitatively characterize the effect of the current density on CF surface wettability, DCA tests were conducted using water and ethylene glycol, and the results are depicted in Figure 6a. The surface free energy of CF, which is divided into dispersive free energy (γ^d^) and polar free energy (γ^p^), was calculated based on the OWRK method, as shown in Figure 6b. As the current density increased, the contact angle between the CF and water first decreased and then increased, whereas the polar free energy showed the opposite trend. Notably, CF with the current density of 0.15 mA/cm^2^ showed the minimum contact angle with water, decreasing from 89.3° to 82.2°. This corresponds to the maximum polar surface free energy of 4.7 mN/m, which is 4.8 times higher than that of the untreated CF. As regards the dispersive component, Figure 6b indicates a completely opposite trend to that of the polar component. As the current density increased, the number of polar functional groups, such as –OH and –C=O, significantly increased (Table 4), causing an increase in the polar free energy. When the current density exceeded 0.15 mA/cm^2^, the polar free energy decreased slightly from 4.7 mN/m to 4.1 mN/m, while the dispersion free energy showed a slight increasing trend, owing to the reduced polar functional group content and electrochemical etching (Figure 3). The greater the polar free energy of the CF surface, the more favorable it is for the resin to fully wet the CF surface, thereby contributing to the improvement in the interfacial bonding strength [30].

### 3.4. Monofilament Tensile Strength

The strength of CFs, as reinforcements in composites, has a significant impact on the mechanical properties of composites. To investigate the influence of current density on the tensile properties of CFs, the monofilament tensile test was performed. Table 6 and Figure 7 present the results and Weibull distributions, respectively. As evident from Table 5, the strength of CF-0 was 5755 MPa. Meanwhile, the tensile strengths under different current densities were 5698, 5704, 5736, and 5743 MPa, which were essentially consistent. Although Figure 3e shows that the surface of CF-4 was slightly etched, the chemical etching effect did not degrade its tensile strength. The results of the monofilament tensile test indicated that the current densities in this experiment did not cause damage to the tensile strength of the CFs.

### 3.5. IFSS of CF/EP

To further analyze the correlation between the current density and interfacial strength of the CFRPs, microsphere debonding tests, for determining the interfacial strength at microscale, were performed on the CF samples treated at different current densities. The IFSS improvements were thus characterized and the results are shown in Figure 8a. As evident from Figure 8a, the IFSS between CF-0 and EP is the lowest (86.2 MPa), owing to the inactive surface of CF-0. As the current density increased, the IFSS showed an upward trend, which is consistent with the chemical states of the CF surfaces. The highest IFSS value (128.3 MPa) was attained at 0.15 mA/cm^2^ and was 48.8% higher than the IFSS of CF-0/EP. As the current density continued to increase, the CF-4/EP IFSS (125.1 MPa) showed a consistent value with CF-3/EP. Considering the results of the surface morphology (Figure 4), chemical characteristics (Table 4), and wettability (Figure 6) of the CFs treated at different current densities, the IFSS variation can be analyzed from the following perspectives. First, there was no significant change in the groove shape and roughness of the CF surfaces, indicating that the electrochemical treatment at various current densities had little effect on the physical interlocking between CF and EP. Second, the electrochemical treatment increased the number of active functional groups on the CF surfaces to varying degrees, effectively improving the polar surface free energy and facilitating the wetting between the EP and CFs. Simultaneously, active functional groups, including –OH, –C=O, –COOH, and –NH_2_, on the CF surfaces were likely involved in the EP curing process, which helps to establish a chemical bonding between CF and EP. Consequently, the CF/EP IFSS significantly improved as the current density increased from 0 to 0.15 mA/cm^2^. Although excessive electrochemical treatment at 0.20 mA/cm^2^ caused slight oxidation etching, which led to a decrease in the number of active functional groups, the IFSS of CF-4/EP did not show significant reduction.

Figure 8b,c show the SEM images of CF-0 and CF-3 after the microsphere debonding tests. Evidently, the morphology of the sample images shows remarkable variation. In Figure 8b, no residual EP resin in the debonding area is observed for CF-0, while the groove morphology of the CF surface in the testing area is clearly discernible. It can be inferred that the adhesion between the CF-0—electrochemically treated at 0 mA/cm^2^, with few active functional groups—and EP resin relied solely on mechanical interlocking and weak van der Waals force, resulting in a lower IFSS. By contrast, in the case of CF-3, a large amount of residual EP resin fragments remained in the debonding area, and the uneven CF surface was covered with a layer of resin, indicating that the bonding strength between the CF-3 and EP resin was evidently improved because of more active functional groups and better wettability due to the electrochemical treatment. This finding is also consistent with the IFSS results.

### 3.6. Mechanical Properties of CF/EP

Figure 9 shows the impact load–time and impact load–displacement curves of the CF/EP composite samples under different current intensities during the impact process, and the results of the compressive strength after the impact test are presented in Table 7 and Figure 10. The damaged area of the composite samples after the impact decreased by 61%, from 732 mm^2^ to 286 mm^2^ (0.15 mA/cm^2^). The compressive strength after impact increased by 28.8%, from 257 MPa to 331 MPa (0.10 mA/cm^2^). Following the electrochemical treatment of the CFs, the impact damage resistance and toughness of the composites significantly increased.

As shown in Figure 9, the local peaks or inflection points on the load–time curves correspond to the internal damage process of the laminates. Schoeppner et al. [10] found that impacted laminates can absorb energy through different forms of damage, including surface pits (plastic deformation), interlayer delamination, and back-splitting, corresponding to microscale EP resin cracks, CF fractures, and CF/EP debonding, respectively, among which interlayer delamination is determined by the interfacial strength of the CF/EP. As the degree of electrochemical treatment increased, the maximum force experienced by the composite samples during the impact process and the pit depth remained similar. However, the number of load drop points on the curves gradually decreased, indicating a reduction in the interlayer delamination within the laminates [18]. Concurrently, the load drop on the curves also decreased as the damage area decreased gradually, which implies that, even if delamination occurred within the laminates, the crack propagation process between the layers was quickly suppressed, avoiding the rapid expansion of delamination. Therefore, the improvement in the CF/EP interface performance due to the increased degree of electrochemical treatment significantly enhanced the impact delamination resistance of the CF/EP composites.

During the impact process, part of the impact energy (E_I_) was stored as elastic potential energy within the laminates, and this process was reversible. However, the remaining energy was absorbed by the sample, referred to as the absorbed energy (E_ab_). Integrating the enclosed area of the load–displacement curves can yield E_ab_ during the impact process of the laminates. From Figure 11, the E_ab_ results indicate that the energy absorbed by each group of samples during the impact process was essentially the same.

From an energy perspective, samples with a higher current intensity have a stronger interface strength, and more energy is consumed for CF/EP debonding and CF pull out, limiting the delamination damage to a smaller area during the impact process (Figure 10). However, excessive electrochemical treatment leads to strong CF/EP interfacial adhesion, limiting the energy dissipation during impact and causing the premature brittle fracture of the CF [11]. As indicated by the nondestructive testing (Figure 10), internal damage of the laminates (Figure 12a), and damage to the back of the sample (Figure 12b), enhanced electrochemical treatment leads to a gradual decrease in the internal damage area of the laminates, and the damage mode gradually changes from interlayer delamination to matrix cracking and fiber fracture.

As regards the CF/EP composites with a low interfacial strength under the weak electrochemical treatment, the laminates suffered severe delamination damage after impact. Extensive areas of separated sublayers formed within the laminates, with the loss of support and protection from the adjacent layers. When compressed, buckling deformation easily occurred, resulting in the rapid failure of the samples and the low compressive strength after impact [9]. With enhanced electrochemical treatment, the CF/EP interface bonding increased, resulting in improved impact delamination resistance and increased undamaged area capable of withstanding high compression loads [21]. Simultaneously, the high interlayer strength can hinder the further expansion of open cracks during compression, thereby effectively improving the compressive strength of the laminates after impact. For the CF-3/EP and CF-4/EP composites, interlayer delamination was scarce and limited to an extremely small range. However, the nondestructive test images of the samples after impact show that the damaged area changed in shape from circular to irregular, indicating the occurrence of CF fracture [11]. Because the electrochemical treatment used in this study did not deteriorate the CF mechanical properties, the reduced load-bearing capacity of the laminates after impact can be attributed to internal fiber fracture.

Although the CF-3/EP samples have the smallest damage area (286 mm^2^), the static mechanical properties of samples decrease slightly owing to fiber fracture. Relatively, the CAI of CF-2/EP reached the highest value (331 MPa) with a larger damage area (352 mm^2^) than CF-3/EP. In summary, 0.10 mA/cm^2^ is more inclined to be used as the current density of electrochemical treatment in engineering.

From Table 8, it can be seen that the optimized CF-2/EP composite has a comparable compressive strength after impact to the well-known T800-grade carbon fiber-reinforced epoxy resin composites from Toray and Hexcel, and has reached the international advanced level.

## 4. Conclusions

This study demonstrated a fast and efficient approach to benefit the interfacial bonding and toughness of CFRP via continuous electrochemical oxidation. By characterizing the surface morphology and chemical activity of CF treated by different current densities, it can be seen that appropriate electrochemical oxidation treatment helps to enhance the surface activity of CF and strengthen the interfacial bonding between CF and EP, which increased the CF/EP IFSS from 86.2 to 128.3 MPa, by 48.8%. Accordingly, the delamination of the CF/EP laminates after impact was significantly reduced, and the impact resistance of the composites was remarkably improved, leading to a significant decrease in the damage area of the laminates after impact by 61%. Moreover, the main energy-dissipation mode changed from interlayer delamination to fiber fracture and resin failure, leveraging the fiber reinforcement effect. However, an excessively strong interfacial strength can cause fiber breakage during impact, reducing the CAI from 331 to 317 MPa, thereby affecting the mechanical properties of the composites. Thus, the optimal current intensity for electrochemical oxidation is deemed to be 0.10 mA/cm^2^, which shows the highest CAI of 331 MPa, reaching the international advanced level.

## Figures and Tables

**Figure 1 polymers-17-01007-f001:**
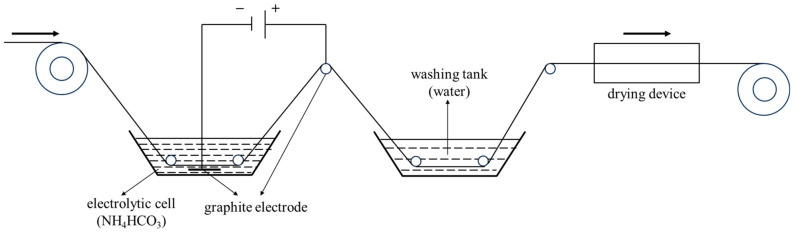
Schematic of continuous electrochemical treatment.

**Figure 2 polymers-17-01007-f002:**
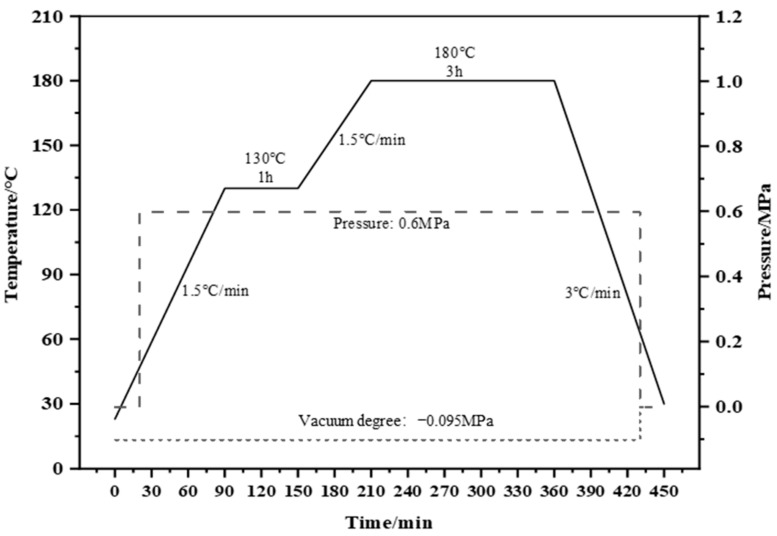
Curing cycle of the epoxy used in this study.

**Figure 3 polymers-17-01007-f003:**
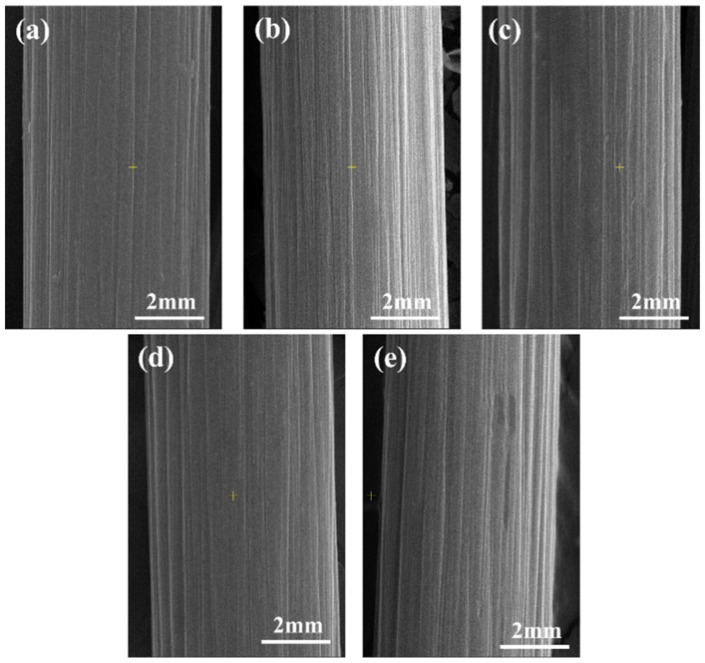
SEM images of the CF surfaces with different current densities: (**a**) CF-0; (**b**) CF-1; (**c**) CF-2; (**d**) CF-3; (**e**) CF-4.

**Figure 4 polymers-17-01007-f004:**
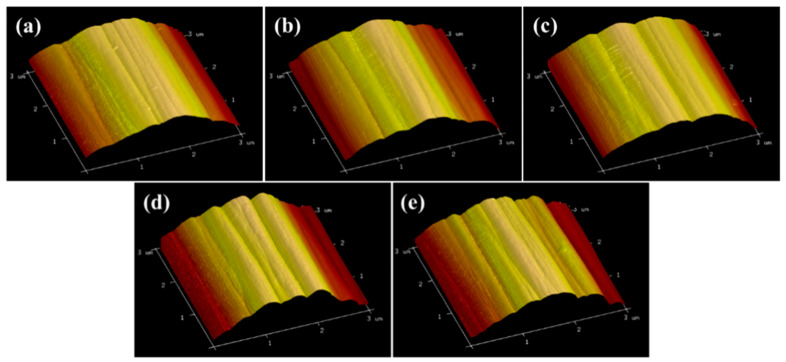
AFM images of the CF surfaces with different current densities: (**a**) CF-0; (**b**) CF-1; (**c**) CF-2; (**d**) CF-3; (**e**) CF-4.

**Figure 5 polymers-17-01007-f005:**
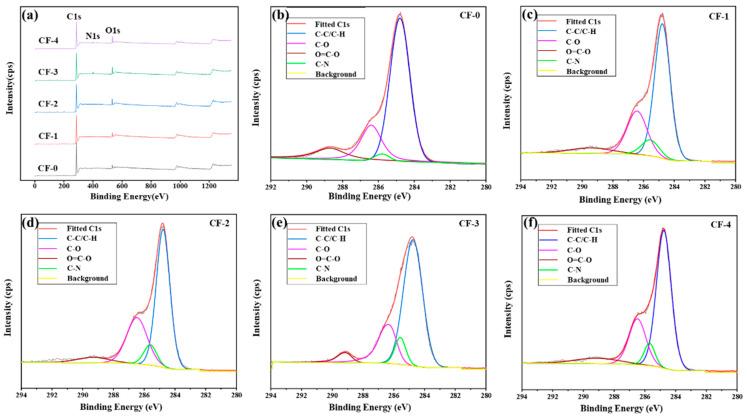
(**a**) Wide-scan spectra of the CF surfaces, showing the high-resolution C1s spectra: (**b**) CF-0; (**c**) CF-1; (**d**) CF-2; (**e**) CF-3; (**f**) CF-4.

**Figure 6 polymers-17-01007-f006:**
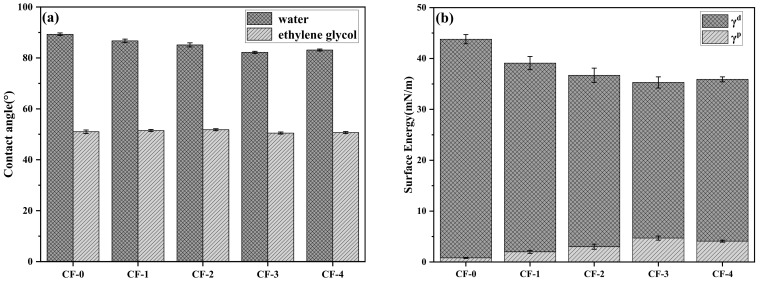
(**a**) Dynamic contact angles and (**b**) surface energies of the CFs.

**Figure 7 polymers-17-01007-f007:**
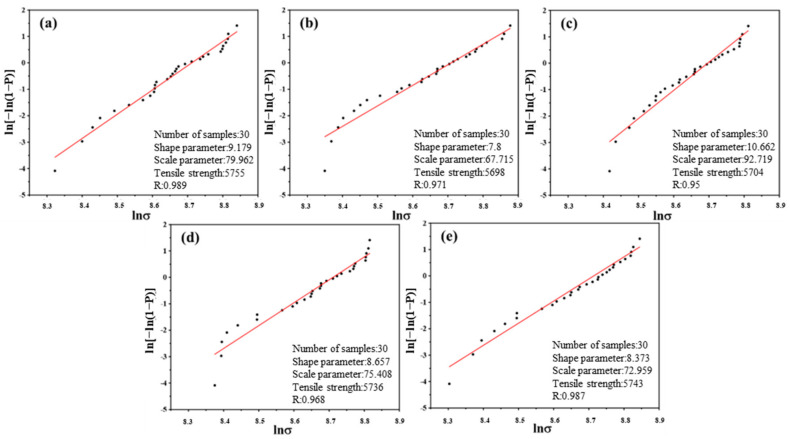
Monofilament strength fitting curves using Weibull distribution: (**a**) CF-0; (**b**) CF-1; (**c**) CF-2; (**d**) CF-3; (**e**) CF-4.

**Figure 8 polymers-17-01007-f008:**
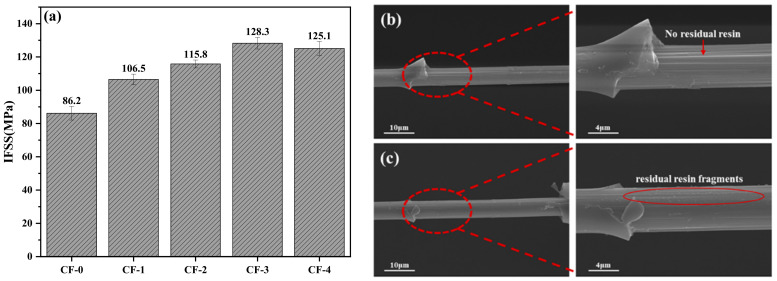
(**a**) IFSS of CF/EP. CF surface morphology after debonding: (**b**) CF-0 and (**c**) CF-3.

**Figure 9 polymers-17-01007-f009:**
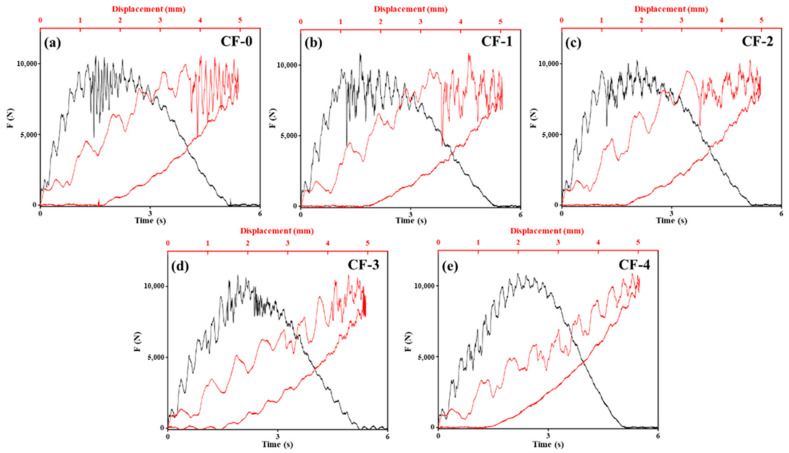
Impact load–time curves of CF/EP composites: (**a**) CF-0; (**b**) CF-1; (**c**) CF-2; (**d**) CF-3; (**e**) CF-4.

**Figure 10 polymers-17-01007-f010:**
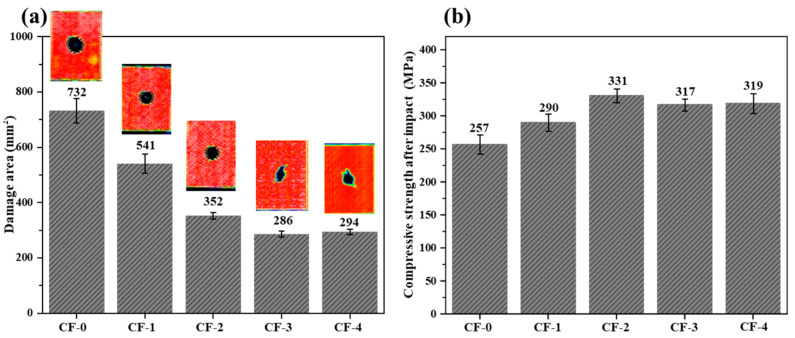
(**a**) Damage area of the samples after impact, along with nondestructive testing images; (**b**) compression strengths after impact of the CF/EP composites.

**Figure 11 polymers-17-01007-f011:**
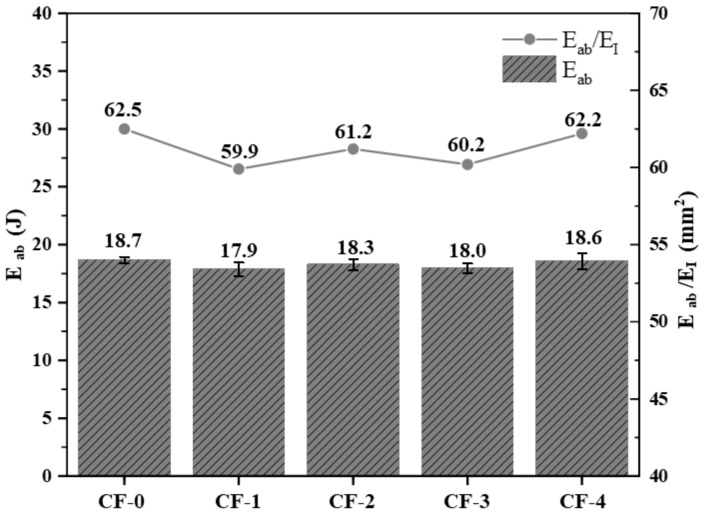
Energy absorption of the CF/EP samples.

**Figure 12 polymers-17-01007-f012:**
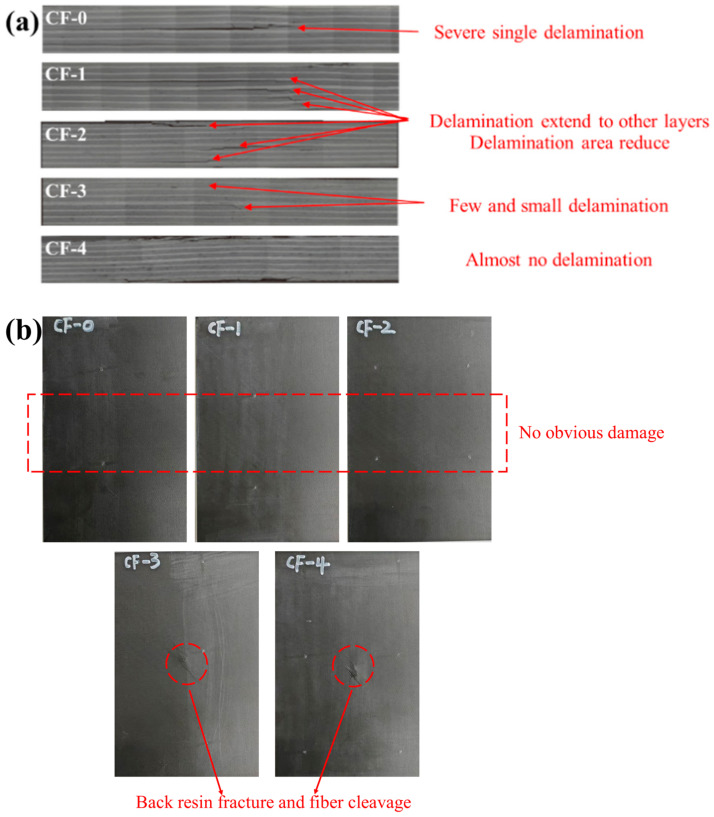
(**a**) Damage on the back of the series CF/EP samples after impact; (**b**) internal damage of the series CF/EP samples after impact.

**Table 1 polymers-17-01007-t001:** Basic properties of the CF used in this study.

Properties	Tensile Strength (MPa)	Elastic Modulus (GPa)	Elongation (%)
Value	5693	293	1.89

**Table 2 polymers-17-01007-t002:** Information of the CF samples.

Samples	CF Type	Continuous Electrochemical Treatment Parameters			
		Current Density	Temperature	Time	Electrolyte
CF-0	T800H grade	0 mA/cm^2^	30 °C	70 s	NH_4_HCO_3_ solution (10 wt%)
CF-1		0.05 mA/cm^2^			
CF-2		0.10 mA/cm^2^			
CF-3		0.15 mA/cm^2^			
CF-4		0.20 mA/cm^2^			

**Table 3 polymers-17-01007-t003:** Surface roughnesses of the CF samples.

Samples	CF-0	CF-1	CF-2	CF-3	CF-4
R_q_/nm	33.6 ± 3.5	31.6 ± 4.0	34.5 ± 5.5	33.1 ± 6.2	32.9 ± 5.1
R_a_/nm	29.1 ± 4.2	26.6 ± 4.6	27.4 ± 4.8	25.4 ± 4.1	25.0 ± 3.9

**Table 4 polymers-17-01007-t004:** Elemental compositions of the CF surfaces.

Sample	Relative Content of Element (%)			O/C	N/C
	C1s	O1s	N1s		
CF-0	91.41	7.66	0.93	0.084	0.010
CF-1	87.58	10.73	1.69	0.123	0.019
CF-2	84.06	13.07	2.87	0.155	0.034
CF-3	82.76	14.35	2.89	0.173	0.035
CF-4	83.16	13.78	3.06	0.166	0.037

**Table 5 polymers-17-01007-t005:** High-resolution XPS results of C1s.

Samples	–C–C–/–C–H		–C–O–		–C=O		–C–N–		Active Functional Group Ratio (%)
	B.E. (eV)	P.C. (%)	B.E. (eV)	P.C. (%)	B.E. (eV)	P.C. (%)	B.E. (eV)	P.C. (%)	
CF-0	284.8	78.15	286.4	15.37	288.7	5.25	285.8	1.23	21.85
CF-1	284.8	69.92	286.5	21.36	288.8	6.23	285.7	2.49	30.08
CF-2	284.7	67.03	286.4	23.10	288.7	6.53	285.7	3.34	32.97
CF-3	284.8	65.80	286.5	24.14	288.6	6.48	285.7	3.58	34.20
CF-4	284.6	66.62	286.3	21.84	288.7	7.41	285.8	4.13	33.38

**Table 6 polymers-17-01007-t006:** Monofilament tensile results of the CFs.

Carbon Fiber	Number of Samples	Shape Parameter	Scale Parameter	Tensile Strength/MPa	R
CF-0	30	9.179	79.96	5755	0.989
CF-1	30	7.800	67.72	5698	0.971
CF-2	30	10.66	92.72	5704	0.950
CF-3	30	8.657	75.41	5736	0.968
CF-4	30	8.373	72.96	5743	0.987

**Table 7 polymers-17-01007-t007:** Results of the impact test of the CF/EP composites under different current intensities.

Sample	Peak Load (kN)	Pit Depth (mm)
CF-0/EP	10.6 ± 0.06	0.20 ± 0.008
CF-1/EP	10.8 ± 0.09	0.21 ± 0.005
CF-2/EP	10.3 ± 0.10	0.22 ± 0.008
CF-3/EP	10.9 ± 0.02	0.20 ± 0.010
CF-4/EP	10.8 ± 0.06	0.22 ± 0.009

**Table 8 polymers-17-01007-t008:** Comparison of the compressive strength after impact with publicly available composites.

Brand	CF-2/EP	M21/IM7	3900-2/T800H	M21E/IMA	X850/IM8
CAI (MPa)	331	273	315	334	330

## Data Availability

Data are contained within the article.

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
