# Peer review of "Synergetic Improvement of Interfacial Performance and Impact Resistance of Carbon Fiber-Reinforced Epoxy Composite via Continuous Electrochemical Oxidation"

_polymers, 2025, doi:10.3390/polym17081007_

Round 1
Reviewer 1 Report
Comments and Suggestions for Authors
The article is devoted to the study of the impact strength of epoxy composites modified with carbon nanofibers using the electrochemical oxidation method.
The article is original and of interest to specialists.
However, the article requires significant revision.
The Introduction needs to be revised.
A large part of the last paragraph (devoted to research methods) should be shortened, as it essentially duplicates information from Section 2, "Materials and Methods."
In general, the last paragraph should reflect the problem statement as a logical conclusion from the literature review.
Section 2.2 would benefit from the addition of a table describing the samples.
The paragraph before Section 3.1 should be removed. It is likely a remnant from the article template.
Figure 6b. The error in determining the surface energy must be indicated.
The dynamic contact angle (considering measurement errors) does not change.
Therefore, the authors need to revise this section, taking into account the measurement errors of the discussed quantities (dynamic contact angle, surface energy).
Table 5. The measurement errors must be indicated.
Figure 8a. Considering the measurement errors, the properties for samples CF1-CF4 are identical. The authors should rewrite the discussion of these results, taking into account the measurement errors.
Figure 9 should logically be moved to the "Data in Brief" section. The data in Figures 10 and 11 are sufficient.
Table 6 is also redundant and duplicates the data from Figure 11.
Figure 12b, images of samples CF3 and CF4. The reviewer could not discern any changes. The images should be revised (at least by adjusting the contrast).
The Conclusions need to be revised in light of the previous comments.
Reviewer 2 Report
Comments and Suggestions for Authors
A manuscript entitled “Synergetic improvement of interfacial performance and impact resistance of carbon fiber reinforced epoxy composite via continuous electrochemical oxidation” needed some modifications. The changes required in the manuscript is as follow:
- Author must incorporate need/motivation (2-3 sentences) to attract the attraction of readers before explaining their work.
- Authors have written well structured and excellent introduction part, but author needs to rewrite the novelty part because it is lagging in continuity and finding novelty
- How electrolysis temperature and time were 30 °C and 70 s respectively, along with current densities considered were 0, 0.05, 0.10, 0.15, and 0.20 mA/cm2, selected by authors? Taken from literature give proper citation or any other source.
- Author needs to mention different standard followed for different properties along with other precautions taken in section 2 at appropriate places.
- The text written in Figure 5, Figure 7 and Figure 9 are not readable. Do the needful changes to make it readable.
- Author needs to add the micrographs of contact angle of all compositions shown in Figure 6a.
- Author also provides the full procedure followed to find surface energies of CFs as supplementary material.
- SEM micrograph scales in Figure 8 b and c not visible. Redraw the scale.
- Author provides the calculations used to plot Figure 11. Specifically, how Ei, Eab, and their ratio calculated in supplementary file.
- Author needs to add one comparison table, in which they compare their results with open literature, quantitively.
- Rewrite the conclusions precisely by removing repetition as discussed in abstract and discussion part. Only give the crux/key outcomes having significance for scientific community along with future road map.
- Author needs to modified references, as references are out-dated. At least 50% of references must be from last five years (2021-2025)
Round 2
Reviewer 1 Report
Comments and Suggestions for Authors
The article can be published.